# A positive feedback loop between Myc and aerobic glycolysis sustains tumor growth in a *Drosophila* tumor model

Kenneth Kin Lam Wong[1,2], Jenny Zhe Liao[1], Esther M Verheyen[1,2]*

[1]Department of Molecular Biology and Biochemistry, Simon Fraser University, Burnaby, Canada; [2]Centre for Cell Biology Development and Disease, Simon Fraser University, Burnaby, Canada

**Abstract** Cancer cells usually exhibit aberrant cell signaling and metabolic reprogramming. However, mechanisms of crosstalk between these processes remain elusive. Here, we show that in an in vivo tumor model expressing oncogenic *Drosophila* Homeodomain-interacting protein kinase (Hipk), tumor cells display elevated aerobic glycolysis. Mechanistically, elevated Hipk drives transcriptional upregulation of *Drosophila* Myc (dMyc; MYC in vertebrates) likely through convergence of multiple perturbed signaling cascades. dMyc induces robust expression of *pfk2* (encoding 6-Phosphofructo-2-kinase/fructose-2,6-bisphosphatase; PFKFB in vertebrates) among other glycolytic genes. Pfk2 catalyzes the synthesis of fructose-2,6-bisphosphate, which acts as a potent allosteric activator of Phosphofructokinase (Pfk) and thus stimulates glycolysis. Pfk2 and Pfk in turn are required to sustain dMyc protein accumulation post-transcriptionally, establishing a positive feedback loop. Disruption of the loop abrogates tumorous growth. Together, our study demonstrates a reciprocal stimulation of Myc and aerobic glycolysis and identifies the Pfk2-Pfk governed committed step of glycolysis as a metabolic vulnerability during tumorigenesis.
DOI: https://doi.org/10.7554/eLife.46315.001

*For correspondence:
everheye@sfu.ca

**Competing interests:** The authors declare that no competing interests exist.

## Introduction

In the 1920s, Otto Warburg first discovered that cancer cells vigorously take up glucose and preferentially produce lactate even in the presence of oxygen, a phenomenon now widely termed the Warburg effect or aerobic glycolysis (*Warburg et al., 1927*). Despite his pioneering work, the Warburg effect was largely disregarded for the subsequent decades (*Liberti and Locasale, 2016*; *Koppenol et al., 2011*). After the discoveries of oncogenes and tumor suppressor genes, cancers are generally considered as genetic diseases rather than metabolic ones (*Wishart, 2015*; *Seyfried et al., 2014*). Not until the 1980s did the revisiting of the Warburg effect in connection with oncogenes spark substantial research in cancer metabolism (*Koppenol et al., 2011*), laying the foundation for 2-deoxy-2-($^{18}$F)fluoro-D-glucose ($^{18}$F-FDG) positron emission tomography (PET) in clinical cancer diagnosis (*Vander Heiden et al., 2009*; *Ben-Haim and Ell, 2008*), recognition of metabolic reprogramming as a hallmark of cancer (*Hanahan and Weinberg, 2011*; *Ward and Thompson, 2012*) and development of anti-cancer agents targeting aerobic glycolysis (*Ganapathy-Kanniappan and Geschwind, 2013*; *Pelicano et al., 2006*; *Granchi and Minutolo, 2012*).

How the Warburg effect arises and contributes to tumor progression have always been the center in the field of cancer metabolism (*Lu et al., 2015*). Warburg hypothesized that mitochondrial impairment is the cause of aerobic glycolysis and cancer (*Warburg, 1956*), which has sparked much controversy among scientists (*Senyilmaz and Teleman, 2015*). Contrary to his idea, the current, widely-accepted view is that oncogenic drivers such as RAS, MYC, hypoxia-inducible factors (HIFs) and steroid receptor coactivators (SRCs) promote cancer cell proliferation and directly stimulate

**eLife digest** Cancer arises when cells in the body divide and grow excessively. These cells will often also take up more glucose than normal cells and break it down into another chemical known as lactate faster. This change to the chemical reactions happening within the cell, also called a metabolic change, is required to help produce the extra DNA, proteins and fatty molecules that it needs to grow.

Elevated levels of certain proteins can trigger the changes that lead to the growth of tumors. MYC (called dMyc in fruit flies) is one of these proteins. It controls cell division and increases the production of enzymes that break down glucose. Hipk is another protein that can induce tumor growth in fruit flies, but how it does so was unknown.

Here, Wong et al. show that high levels of Hipk boost glucose metabolism and accumulation of dMyc protein in fruit fly cells. They also describe the link between increased glucose metabolism and uncontrolled cell division.

First, fruit fly cells were fed a fluorescent molecule similar to glucose that cannot be broken down by the cells. This experiment established that glucose uptake increases in cells with too much Hipk. These cells also break down glucose faster, confirming that they have increased glucose metabolism.

Cells with high levels of Hipk also activate the genes that generate the enzymes involved in glucose breakdown, and increase the activity of the gene coding for dMyc. Levels of the dMyc protein are higher in these cells, which was shown by staining the cells with fluorescent molecules that specifically bind the dMyc protein. It is this buildup of dMyc protein that activates the genes coding for the enzymes responsible for glucose breakdown. PFK2 is one of these enzymes.

Finally, Wong et al. inhibited the production of the enzymes that are elevated in cells with high Hipk. Stopping the production of PFK2 prevents both tumor growth and the accumulation of dMyc protein. This shows that high levels of dMyc increase PFK2 levels, leading to further dMyc buildup, and creating a loop that links the uncontrolled cell division caused by too much dMyc and the shift to higher glucose metabolism.

These results highlight new potential targets for cancer therapy, showing that targeting glucose metabolism may reduce, or even stop, tumor growth.

DOI: https://doi.org/10.7554/eLife.46315.002

aerobic glycolysis through regulating transcriptional expression or catalytic activities of metabolic enzymes (*Koppenol et al., 2011*). Several explanations for the Warburg effect have been put forward, including rapid adenosine triphosphate (ATP) production and de novo biosynthesis of macromolecules (*Liberti and Locasale, 2016*). Recently, Warburg effect functions are being re-evaluated as a growing body of evidence shows that metabolic rewiring in cancers impacts cell signaling and epigenetics (*Liberti and Locasale, 2016*; *Lu and Thompson, 2012*).

*Drosophila* has proven to be a powerful genetic model organism for studying tumorigenesis in vivo largely due to high conservation of genes and signaling cascades between human and flies and reduced genetic redundancy (*Gonzalez, 2013*; *Herranz and Cohen, 2017*) (*Supplementary file 1*). In a recent study, we showed that elevation of *Drosophila* Hipk causes an in vivo tumor model characterized by tissue overgrowth, loss of epithelial integrity and invasion-like behaviors (*Blaquiere et al., 2018*). The tumorigenic roles of *Drosophila* Hipk seem to be conserved in mammals as the four members of the HIPK family (HIPK1-4) are also implicated in certain cancers (Reviewed in *Blaquiere and Verheyen, 2017*). For example, HIPK1 is highly expressed in breast cancer cell lines, colorectal cancer samples and oncogenically-transformed mouse embryo fibroblasts (*Kondo et al., 2003*; *Rey et al., 2013*). Also, HIPK2 is elevated in certain cancers including cervical cancers, pilocytic astrocytomas, colorectal cancer cells and in other proliferative diseases, such as thyroid follicular hyperplasia (*Al-Beiti and Lu, 2008*; *Cheng et al., 2012*; *Yu et al., 2009*; *D'Orazi et al., 2006*; *Saul and Schmitz, 2013*; *Deshmukh et al., 2008*; *Lavra et al., 2011*). To gain a better understanding of cancer metabolism, we set out to use the fly Hipk tumor model to investigate whether and how cellular metabolism is altered in tumor cells. We find that Hipk-induced tumorous growth is accompanied by elevated aerobic glycolysis. Furthermore, we identify novel

feedback mechanisms leading to prolonged dMyc expression and hence tumorigenesis. Our study reveals potential metabolic vulnerabilities that could be exploited to suppress tumor growth.

## Results

### Hipk tumor cells exhibit elevated glucose uptake and consumption

Larval imaginal epithelia such as wing and eye-antennal imaginal discs, which give rise to wing and eye structures in adult flies, respectively, are extensively used as tumor models to study human carcinomas (*Herranz et al., 2016*). To generate the Hipk tumor model, we used the Gal4-UAS system (*Brand and Perrimon, 1993*) to induce overexpression of *hipk* in larval wing discs (full genotype: *dpp-Gal4>UAS-RFP + UAS-hipk*, abbreviated as *dpp>RFP + hipk*) (*Figure 1c–d*). Fluorescent proteins, for example red fluorescent proteins (RFP) or green fluorescent proteins (GFP), were co-expressed to label the transgene-expressing cells. As previously reported (*Blaquiere et al., 2018*), in contrast to control discs (*dpp>RFP*) (*Figure 1a–b*), tumorous growth and severe tissue distortions were evident in *hipk*-expressing discs (*dpp>RFP + hipk*) (*Figure 1c–d*).

To ask if elevated Hipk alters cellular metabolism, we first used a fluorescently labeled glucose analog, 2-deoxy-2-[(7-nitro-2,1,3-benzoxadiazol-4-yl)amino]-D-glucose (2-NBDG) (*Figure 1j*). Like D-glucose, 2-NBDG is transported into cells through glucose transporters and phosphorylated by hexokinases (*Yoshioka et al., 1996*; *O'Neil et al., 2005*). However, due to the presence of the fluorescent amino group at the C-2 position, 2-NBDG-6-phosphate cannot proceed further through glycolysis and is consequently trapped within the cells, rendering it a probe for monitoring glucose uptake (*O'Neil et al., 2005*). 2-NBDG is often used in *Drosophila* to measure glucose uptake. For example, overexpression of *glut1* (encoding Glucose transporter 1) is sufficient to promote the accumulation in 2-NBDG (*Niccoli et al., 2016*). Increased import of 2-NBDG is also found in *dMyc*-expressing cells (*de la Cova et al., 2014*) and *Ras$^{V12}$scrib$^{-/-}$* tumor cells (*Katheder et al., 2017*), indicating that such cells acquire enhanced glucose metabolism.

Similarly, we found that cells with elevated Hipk (RFP positive) in *hipk*-expressing wing discs (*dpp>RFP + hipk*) exhibited statistically significant 2-NBDG accumulation when compared with the neighboring wild-type cells (RFP negative) (*Figure 1c–d and i*) or cells (either RFP positive or negative) in control discs (*dpp>RFP*) (*Figure 1a–b*). A comparable phenomenon was seen in larval eye-antennal discs (*Figure 1—figure supplement 1a–b*). Furthermore, we detected a mild upregulation of *glut1* (1.6-fold) in *hipk*-expressing discs by quantitative real-time PCR (qRT-PCR) (*Figure 2a*). Knockdown of *glut1* using RNA interference (RNAi) moderately reduced 2-NBDG incorporation in Hipk tumor cells (*Figure 1—figure supplement 1c–d*). The knockdown efficiency of *glut1-RNAi* was tested using a ubiquitous Gal4 driver, *act5c-Gal4* (*Figure 1—figure supplement 2a*). Together, these results suggest that elevated Hipk facilitates the import of glucose into cells at least in part through augmenting Glut1 expression.

Next, we evaluated the intracellular glucose levels using a FRET (Förster resonance energy transfer)-based glucose sensor, which is composed of a glucose-binding domain (GBD) fused with cyan fluorescent protein (CFP) and yellow fluorescent protein (YFP) (*Volkenhoff et al., 2018*; *Fehr et al., 2003*) (*Figure 1l*). The binding of glucose to GBD induces conformational changes, leading to increased FRET efficiency (the ratio of FRET to CFP). Intriguingly, a lower FRET/CFP ratio indicative of reduced glucose concentration was observed in *hipk*-expressing cells (*Figure 1g–h and k*) when compared with control discs (*Figure 1e–f and k*). This implies that despite enhanced glucose uptake, glucose is substantially utilized in the rapidly proliferating Hipk tumor cells.

### Elevated Hipk induces robust expression of a subset of glycolytic genes

Glycolysis is a catabolic process that breaks down glucose into pyruvate via multiple enzymatic steps. To ask if glycolysis is upregulated, we examined the expression profile of glycolytic genes by qRT-PCR and changes greater than 1.5-fold were considered significant. We isolated RNA from whole wing discs expressing either GFP (control) or *hipk* under the *dpp-Gal4* driver control. qRT-PCR analyses revealed that genes encoding Hexokinases (Hex-A and Hex-C, 2.2-fold and 4.3-fold respectively), Phosphoglucose isomerase (Pgi, 2.4-fold), Phosphofructokinase 2 (Pfk2, also known as Pfrx, 2.7-fold) and Lactate dehydrogenase (Ldh, also known as Impl3, 3.4-fold) are significantly

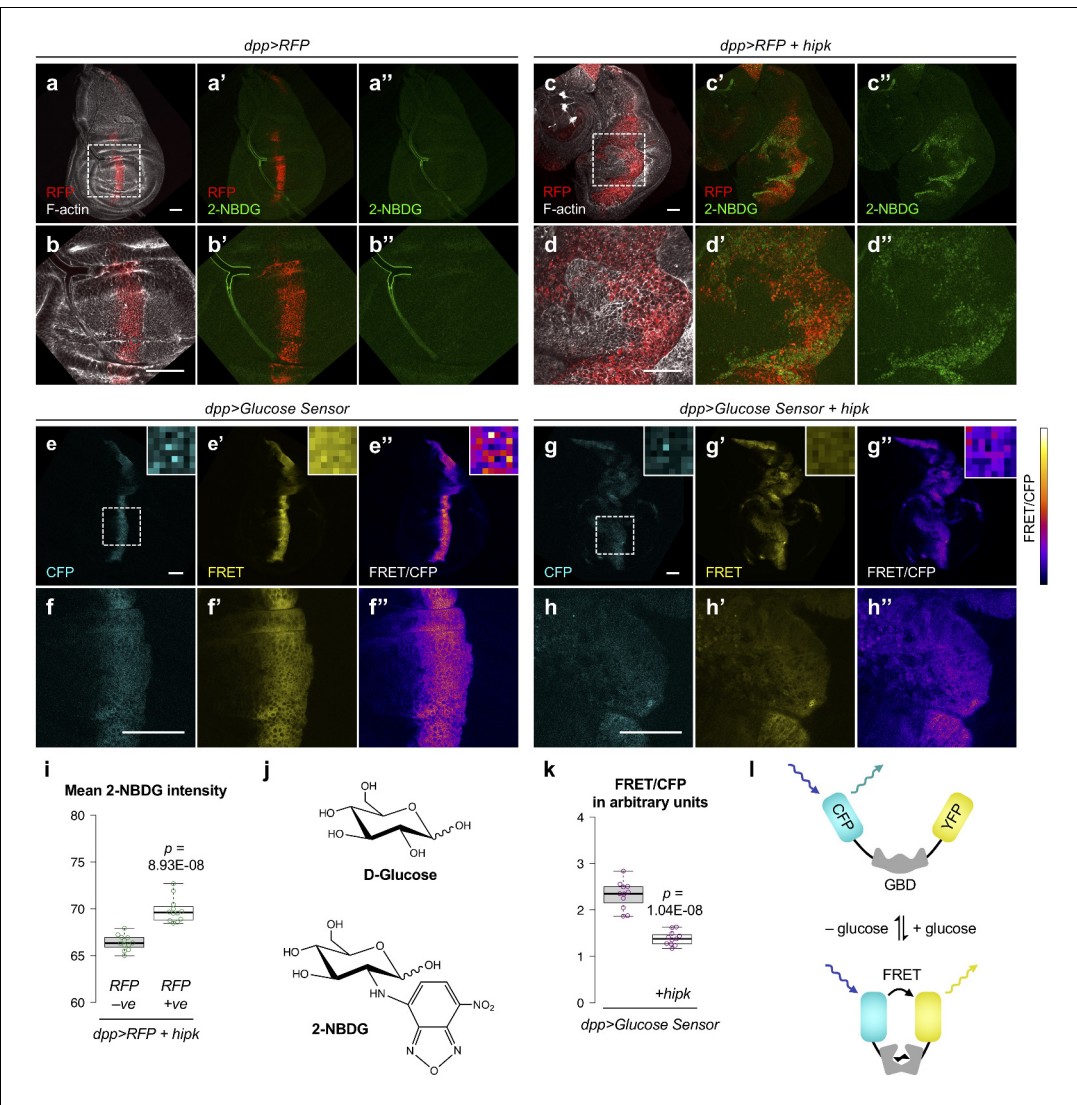

**Figure 1.** Hipk tumor cells exhibit elevated glucose uptake and reduced intracellular glucose levels. (**a–d**) Incorporation of 2-NBDG (green) in control (*dpp>RFP*) (**a**, **b**) and *hipk*-expressing (*dpp>RFP + hipk*) wing discs (**c**, **d**). Insets (dashed line) in (**a** and **c**) are magnified in (**b** and **d**), respectively. RFP (red) marks the transgene-expressing cells. F-actin (gray) staining shows the overall tissue morphology. (**e–h**) Intracellular glucose FRET efficiency (FRET/CFP) (colored using Fire Lookup Table) in control (*dpp>Glucose Sensor*) (**e–f**) and *hipk*-expressing (*dpp>Glucose Sensor+hipk*) wing discs (**g–h**). Individual CFP and FRET channels are shown in cyan and yellow, respectively. Insets (dashed line) in (**e** and **g**) are magnified in (**f** and **h**), respectively. Insets (solid line) at the upper right corners in (**e–e''**) and (**g–g''**) show the corresponding signals at single pixel level. (**i**) Quantification of mean 2-NBDG intensity in *hipk*-expressing cells (RFP positive) and the neighboring wild-type cells (RFP negative) in *hipk*-expressing wing discs (*dpp >RFP + hipk*) ($n = 12$ wing discs). (**j**) Chemical structures of D-glucose (top) and 2-NBDG (bottom). (**k**) Quantification of glucose FRET efficiency in control (*dpp>Glucose Sensor*) and *hipk*-expressing (*dpp >Glucose Sensor + hipk*) wing discs ($n = 11$ wing discs per genotype). (**l**) Schematic diagram of the FRET-based glucose sensor. Glucose binding induces conformational changes of glucose-binding domain (GBD) such that FRET occurs and YFP emission increases. Scale bars, 50 μm. Exact *p* values are shown and calculated using unpaired two-tailed *t*-test.

DOI: https://doi.org/10.7554/eLife.46315.003

The following source data and figure supplements are available for figure 1:

**Source data 1.** Analyses of glucose metabolism in Hipk tumor cells and evaluation of RNAi tools.

DOI: https://doi.org/10.7554/eLife.46315.006

**Figure supplement 1.** Elevated Hipk stimulates 2-NBDG uptake in a Glut1-dependent manner.

DOI: https://doi.org/10.7554/eLife.46315.004

*Figure 1 continued on next page*

*Figure 1 continued*

**Figure supplement 2.** Knockdown efficiencies of the RNAi lines targeting glycolytic genes.
DOI: https://doi.org/10.7554/eLife.46315.005

upregulated in *hipk*-expressing discs (*Figure 2a–b*). Expression of other glycolytic genes, on the other hand, remained relatively unchanged.

As the Dpp domain comprises 10–20% of the wing disc, the glycolytic gene upregulation within *hipk*-expressing cells is likely understated. In light of such a limitation, we used a GFP-based enhancer trap to monitor *Ldh* transcription in vivo (*Ldh-GFP*) (*Quiñones-Coello et al., 2007*). In control wing discs, little *Ldh-GFP* was detected (*Figure 2c*), indicating that *Ldh* is minimally expressed under physiological conditions as previously described (*Wang et al., 2016*). On the contrary, we observed robust *Ldh-GFP* expression in *hipk*-expressing wing discs (*Figure 2d*), confirming that elevated Hipk induces *Ldh* expression.

Hexokinases phosphorylate glucose to form glucose-6-phosphate (G6P), which is the first essentially irreversible step in glycolysis (*Figure 2b*). Pgi catalyzes the reversible conversion of G6P and fructose-6-phosphate. Pfk2 is a bifunctional enzyme that synthesizes fructose-2,6-bisphosphate

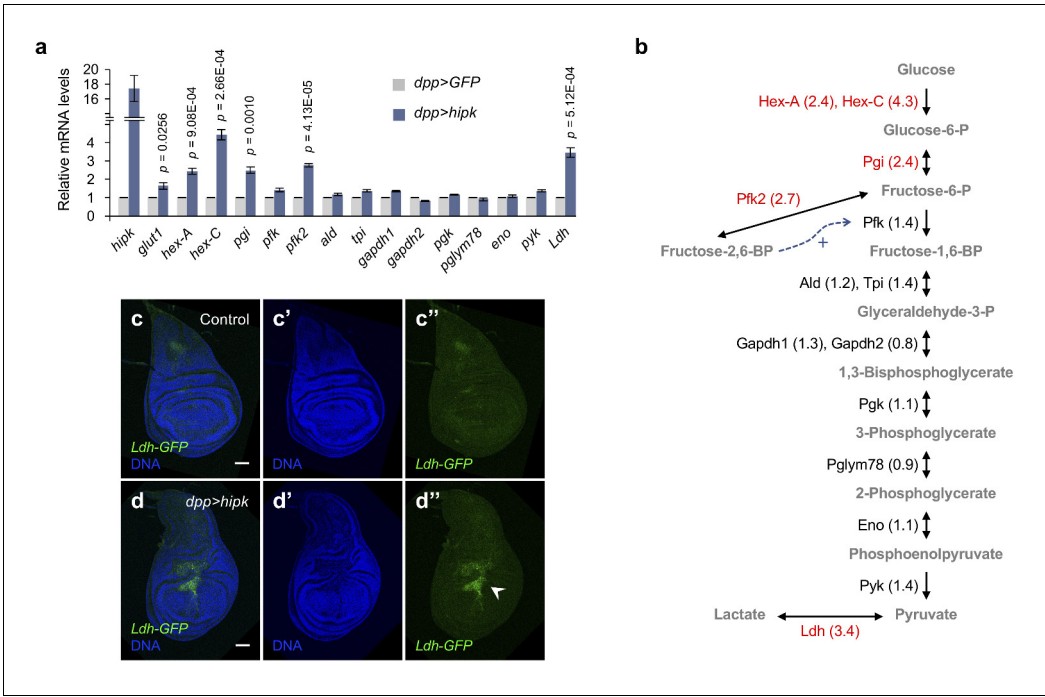

**Figure 2.** Elevated Hipk induces robust expression of a subset of glycolytic genes. (**a, b**) qRT-PCR analysis of glycolytic gene expression in *hipk*-expressing discs (*dpp>hipk*) relative to control discs (*dpp>GFP*) (**a**). Data are mean ± sem. *n* = 3 independent experiments with samples pooled from 10 to 20 wing discs per genotype. Schematic diagram of glycolysis (**b**). Fold changes are shown in brackets. Glycolytic enzymes with more than 1.5-fold transcriptional upregulation in *hipk*-expressing discs are marked in red. Bi- and uni-directional arrows indicate the reversible and essentially irreversible steps in glycolysis, respectively. Dashed arrow (blue) shows the positive allosteric regulation of Pfk by fructose-2,6-BP. Full names of the glycolytic genes are provided in *Supplementary file 1*. Exact *p* values are shown and calculated using unpaired two-tailed *t*-test. (**c–d**) Expression of *Ldh-GFP* (green), an enhancer trap to monitor *Ldh* expression, in control (genotype: *Ldh-GFP/+*) (**c**) and *hipk*-expressing wing discs (genotype: *Ldh-GFP/dpp>hipk*) (**d**). DAPI staining for DNA (blue) reveals tissue morphology. White arrowhead in **d''** indicates marked induction of *Ldh* expression. Scale bars, 50 μm.
DOI: https://doi.org/10.7554/eLife.46315.007

The following source data is available for figure 2:

**Source data 1.** Expression profile of glycolytic genes in*hipk*-expressing discs.
DOI: https://doi.org/10.7554/eLife.46315.008

(F2,6-BP). F2,6-BP is a potent allosteric activator of Pfk, which governs the committed, second irreversible step in glycolysis. Ldh, similar to LDHA in vertebrates (*Rechsteiner, 1970*), favors the reduction of pyruvate to lactate, regenerating nicotinamide adenine dinucleotide (NAD$^+$) such that glycolysis continues unabated (*Valvona et al., 2016*). The robust upregulation of these glycolytic enzymes in *hipk*-expressing discs is in agreement with previous data implying that glucose consumption is stimulated. Taken together, the results support the hypothesis that Hipk tumor cells exhibited elevated glycolytic activities, resembling the Warburg effect seen in cancers.

## Elevated Hipk drives dMyc upregulation in a region-specific manner

To elucidate the underlying mechanisms for the metabolic alterations in the Hipk tumors, we examined gene expression of dMyc and Sima (HIF1-α in vertebrates) since these two transcription factors are well-known inducers of glycolysis in cancer cells (*Miller et al., 2012a*; *Marín-Hernández et al., 2009*). qRT-PCR analyses showed that *dMyc* was upregulated in *hipk*-expressing discs whereas *sima* expression remained unchanged (*Figure 3a*).

Larval wing discs are mainly composed of three regions known as the notum, hinge and pouch, corresponding to the adult structures they give rise to (*Figure 3b*). Using a *dMyc-lacZ* enhancer trap (encoding β-galactosidase (β-gal)) to monitor the transcriptional control of *dMyc* in vivo, we observed that *dMyc* was expressed at high levels in the pouch region of control discs (*Figure 3c*). In the hinge and notum areas of control discs, *dMyc* was minimally and weakly expressed, respectively (*Figure 3c*). In *hipk*-expressing discs, we found intense β-galactosidase staining, confirming that Hipk cell-autonomously drives transcriptional upregulation of dMyc. Interestingly, the increase in the β-gal signal intensity appeared to be the largest in the wing hinge (*Figure 3d*, arrowhead) where the

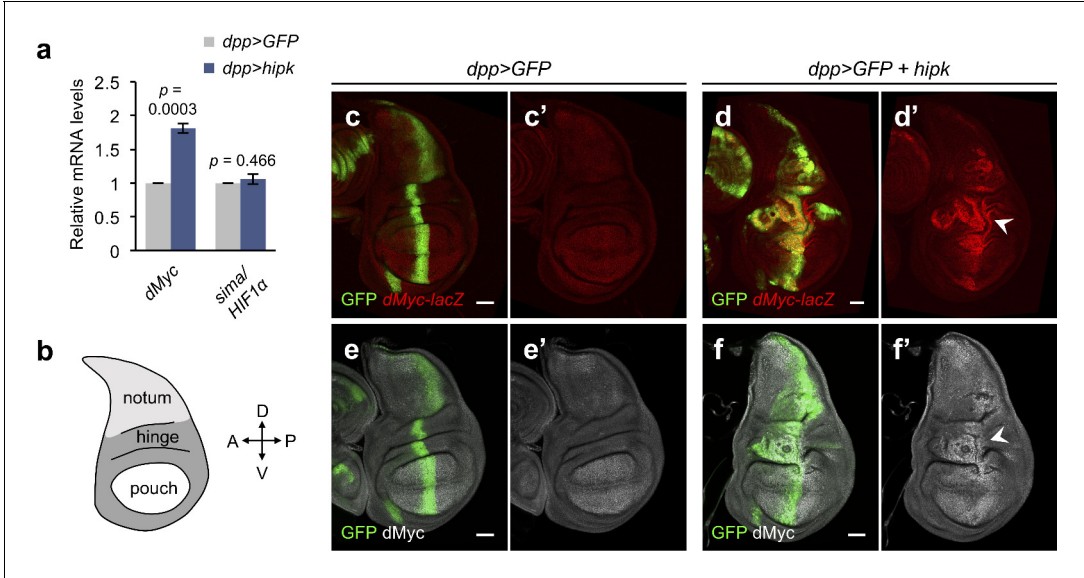

**Figure 3.** Elevated Hipk drives dMyc upregulation in a region-specific manner. (a) qRT-PCR analysis of *dMyc* and *sima* expression in *hipk*-expressing discs (*dpp>hipk*) relative to control discs (*dpp>GFP*). Data are mean ± sem. *n* = 3 independent experiments with samples pooled from 10 to 20 wing discs per genotype. Exact *p* values are shown and calculated using unpaired two-tailed *t*-test. (b) Schematic diagram of a larval wing imaginal disc, with notum, hinge and pouch in light gray, dark gray and white, respectively. (c–d) β-galactosidase staining (red) in control (*dpp>GFP*) (c) and *hipk*-expressing (*dpp>GFP + hipk*) (d) wing discs harboring a *lacZ* reporter to monitor the transcriptional induction of dMyc (*dMyc-lacZ*). White arrowhead in (d′) indicates robust upregulation of *dMyc* in the wing hinge. (e–f) dMyc staining (gray) in control (*dpp>GFP*) (e) and *hipk*-expressing (*dpp>GFP + hipk*) (f) wing discs. White arrowhead in (f′) indicates dMyc accumulation in the wing hinge. GFP (green) marks the transgene-expressing cells. Scale bars, 50 µm.

DOI: https://doi.org/10.7554/eLife.46315.009

The following source data is available for figure 3:

**Source data 1.** Expression levels of *dMyc* and *sima* in *hipk*-expressing discs.
DOI: https://doi.org/10.7554/eLife.46315.010

tumor cells proliferated most rapidly, while the intensity faded when moving away from the hinge region (*Figure 3d*).

Immunofluorescent staining using antibodies against dMyc confirmed the accumulation of dMyc proteins in Hipk tumor cells especially in the wing hinge (*Figure 3e–f*), which positively correlates with the marked increase in *dMyc* mRNA levels in the same region (*Figure 3d*). dMyc proteins appeared to not accumulate in *hipk*-expressing cells in the wing pouch (*Figure 3f*), possibly due to weak induction in dMyc transcription (*Figure 3d*).

The variability in Hipk-induced dMyc accumulation suggests a potential region-specificity in dMyc upregulation. To assess this, we generated random flip-out clones of *hipk*-expressing cells. The clones were marked by RFP. We observed that *hipk*-expressing clones located in the dorsal hinge (*Figure 4c*, arrowhead, *Figure 4—figure supplement 1a*), the lateral hinge (*Figure 4—figure supplement 1b*) as well as the epithelium on the peripodial side exhibited marked dMyc accumulation (*Figure 4—figure supplement 1c*). On the contrary, when *hipk*-expressing clones were located in the wing pouch, dMyc levels appeared unchanged (*Figure 4—figure supplement 1d*). Altogether, our data imply that elevated Hipk drives dMyc upregulation in a region-specific manner.

## Elevated Hipk modulates dMyc expression likely through convergence of multiple perturbed signaling cascades

We further investigated the molecular mechanisms underlying the upregulation of dMyc in Hipk tumor cells. Earlier studies have demonstrated that Hipk is a versatile regulator of numerous signaling pathways. For example, Hipk stimulates the transcriptional activity of Yorkie (Yki, a *Drosophila*

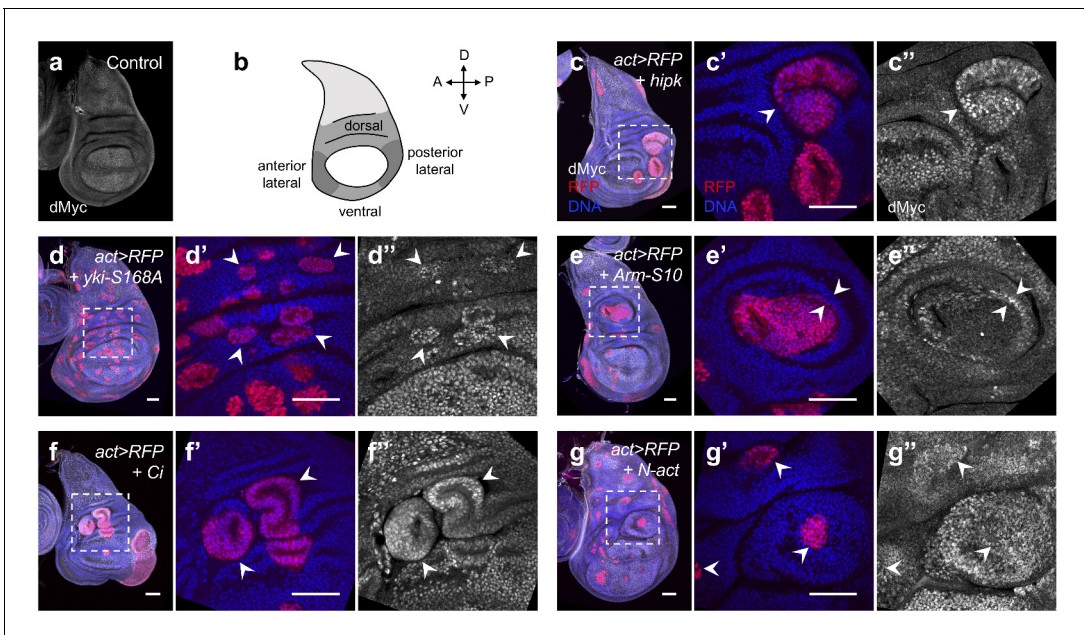

**Figure 4.** Elevated Hipk and perturbations in cell signaling cascades induce dMyc accumulation in the wing hinge region. (**a**) dMyc staining (gray) in a control wing disc. (**b**) Schematic diagram of a larval wing imaginal disc. The hinge region can be divided into dorsal, lateral and ventral compartments. (**c–g**) dMyc staining (gray) in flip-out clones expressing *hipk* (**c**), *yki-S168A* (**d**), *Arm-S10* (**e**), *Ci* (**f**) or *N-act* (**g**) under the control of *actin-Gal4*. RFP (red) marks the transgene-expressing clones. DAPI staining for DNA (blue) reveals tissue morphology. (**c'**, **c''-g'**, **g''**) are magnified images of the insets (dashed lines) in (**c-g**). (**c**) White arrowhead indicates dMyc accumulation in a *hipk*-expressing clone. (**d**) White arrowheads indicate *yki-S168A*-expressing clones with dMyc accumulation. (**e**) White arrowheads indicate ectopic dMyc expression at the boundary of the *Arm-S10*-expressing clone and the neighboring wild-type cells. (**f**) White arrowheads indicate dMyc upregulation in *Ci*-expressing clones. (**g**) White arrowheads indicate dMyc accumulation in *N-act*-expressing clones and the adjacent wild-type cells. Scale bars, 50 μm.

DOI: https://doi.org/10.7554/eLife.46315.011

The following figure supplement is available for figure 4:

**Figure supplement 1.** Effects on dMyc expression by elevated Hipk or perturbations in cell signaling.
DOI: https://doi.org/10.7554/eLife.46315.012

orthologue of YAP/TAZ), the effector of the Hippo tumor suppressor pathway (*Chen and Verheyen, 2012*; *Poon et al., 2012*). Hipk also functions as a positive regulator of both Wnt/Wingless (Wg) and Hedgehog signaling by stabilizing Armadillo (β-catenin in vertebrates) and Cubitus interruptus (Ci; Gli in vertebrates), respectively (*Lee et al., 2009a*; *Swarup and Verheyen, 2011*). In addition, Hipk has been found to antagonize Groucho, a global co-repressor, to promote Notch signal transduction (*Lee et al., 2009b*). If Hipk-induced dMyc upregulation is through deregulation of multiple signaling pathways, we would expect to see dMyc accumulation when the signaling pathways are perturbed. We first focused on the dorsal hinge area since ectopic dMyc expression and tumor growth are most striking in such region in *hipk*-expressing discs.

Flip-out clones expressing a constitutively active form of Yki (Yki-S168A) to inactivate the Hippo pathway were usually round and most of them in the dorsal hinge displayed dMyc upregulation (*Figure 4d*, arrowheads). Wing discs with flip-out clones expressing an active form of Arm (Arm-S10) featured an expansion in the hinge area and some dMyc accumulated at the interface between the clone and the neighboring wild-type cells (*Figure 4e*, arrowheads) but not at the center of the clone. Flip-out clones expressing Ci to activate Hedgehog signaling in the hinge region were round or irregular in shape and exhibited elevated levels of dMyc (*Figure 4f*, arrowheads). Consistent with previous findings (*Djiane et al., 2013*), expression of activated Notch appeared to induce dMyc accumulation in both cell and non-cell autonomous fashions in the hinge area likely involving a signaling relay (*Figure 4g*). Therefore, individual perturbation of signaling pathways that have been previously shown to be regulated by Hipk was sufficient to trigger ectopic dMyc expression in the hinge region.

In the wing pouch region, activation of Yki failed to cause any dMyc upregulation (*Figure 4—figure supplement 1e*). This is in contrast to previous findings that overexpression of Yki increased dMyc protein levels in both hinge and pouch regions (*Ziosi et al., 2010*). Such discrepancy could be due to the different expression levels of the *UAS-Yki* constructs. Flip-out clones expressing Arm-S10 or activated Notch displayed reduced dMyc expression (*Figure 4—figure supplement 1f and h*), which is consistent with earlier studies showing that Wg and Notch signaling negatively regulate dMyc expression in the pouch region (*Duman-Scheel et al., 2004*; *Herranz et al., 2008*). Clones with *Ci* overexpression were few and tiny in the wing pouch and the dMyc levels in the clones seemed comparable to those in the neighboring wild-type cells (*Figure 4—figure supplement 1g*). Our data suggest that the region specificity in dMyc upregulation applies to not only Hipk tumor cells, but also cells with other tumorigenic stimuli.

In summary, our data show that in the wing hinge region where Hipk tumor growth is most prominent, dMyc protein markedly accumulates. Individual perturbations in signaling pathways that have been shown to be regulated by Hipk also result in ectopic dMyc expression in the same region. Hence, it is likely that in the rapidly proliferating Hipk tumor cells, aberrant signals propagate and converge at the transcriptional control of dMyc. In other words, the modulation of dMyc levels by Hipk seems to be a cumulative effect of alterations in multiple signaling outputs.

## Hipk-induced dMyc governs aerobic glycolysis and tumor growth

In addition to growth control, dMyc/MYC plays a conserved role in upregulating glycolysis in vertebrates and *Drosophila* during normal development and in cancers (*de la Cova et al., 2014*; *Gallant, 2013*). As expected and previously reported (*de la Cova et al., 2014*), overexpression of *dMyc* caused marked 2-NBDG uptake (*Figure 5a and d*) and promoted glycolytic gene expression (*Figure 5e*). More importantly, knockdown of *dMyc* by RNAi in *hipk*-expressing discs reversed the 2-NBDG accumulation (*Figure 5b–d*) and moderately suppressed glycolytic gene upregulation (*Figure 5e*), suggesting that dMyc is essential for the enhanced glycolysis in Hipk tumor cells. Additionally, knockdown of *dMyc* significantly suppressed the tumorous growth and tissue distortions in *hipk*-expressing discs, as seen by the decrease in GFP positive cells (*Figure 5f–g*). Using an independent RNAi construct targeting *dMyc* (*dMyc-RNAi #2*), we observed a similar effect upon silencing of *dMyc* (*Figure 5h*). dMyc immunofluorescent staining showed the efficient knockdown when *dMyc-RNAi* constructs were expressed (*Figure 5g–i*). Wing discs with *dMyc* knockdown alone appeared wild-type (*Figure 5i*). Altogether, our results imply that dMyc-induced glycolysis may be a key driver for Hipk tumor growth. Intriguingly, overexpression of *dMyc* alone (*Figure 5a*), despite the enhanced glycolysis, failed to reproduce the tumorous phenotypes observed in *hipk*-expressing discs

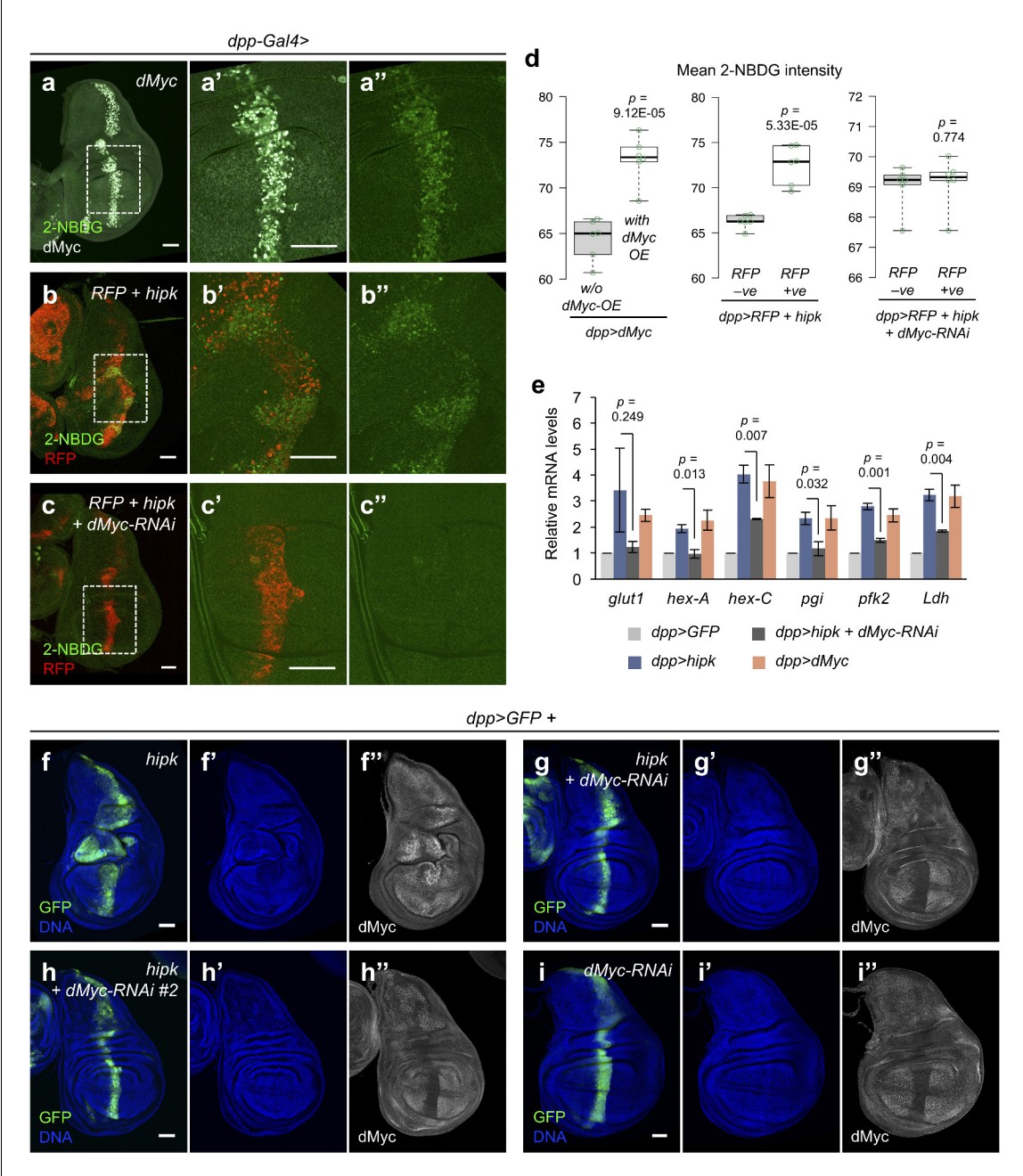

**Figure 5.** dMyc mediates the Hipk-induced glycolysis and tumor growth. (**a–c**) Incorporation of 2-NBDG (green) in *dMyc*-expressing wing disc (*dpp>dMyc*) (**a**), *hipk*-expressing wing disc (*dpp>RFP + hipk*) (**b**), and *hipk*-expressing wing disc with *dMyc* knockdown (*dpp>RFP + hipk + dMyc-RNAi*) (**c**). Insets (dashed line) in (**a, b** and **c**) are magnified in (**a'–a''**), (**b'–b''**) and (**c'–c''**), respectively. dMyc staining (gray) in (**a**) marks the *dMyc*-overexpressing cells. RFP (red) in (**b** and **c**) marks the transgene-expressing cells. (**d**) Quantification of mean 2-NBDG intensity in *dMyc*-expressing cells and the neighboring wild-type cells in *dMyc*-expressing wing discs (*dpp>dMyc*) (left), in *hipk*-expressing cells (RFP positive) and the neighboring wild-type cells (RFP negative) in *hipk*-expressing discs (*dpp>RFP + hipk*) (middle) and, in *hipk, dMyc-RNAi* co-expressing cells (RFP positive) and the neighboring wild-type cells (RFP negative) in *hipk*-expressing discs with *dMyc* knockdown (*dpp>RFP + hipk + dMyc-RNAi*) (right). *n* = 6 discs per genotype. (**e**) qRT-PCR analysis of expression of a set of glycolytic genes in *hipk*-expressing wing discs (*dpp>hipk*), *hipk*-expressing discs with *dMyc* knockdown (*dpp>hipk + dMyc-RNAi*), and *dMyc*-expressing discs relative to control discs (*dpp>GFP*). Data are mean ± sem. *n* = 3 independent experiments with samples pooled from 10 to 20 wing discs per genotype. (**f–i**) dMyc staining (gray in **f''**), (**g''**, **h''** and **i''**) in *hipk*-expressing disc (*dpp>GFP + hipk*) (**f**), *hipk* and *dMyc-RNAi* co-expressing disc (*dpp>GFP + hipk + dMyc-RNAi*) (**g**), *hipk* and *dMyc-RNAi #2* co-expressing disc (*dpp>GFP + hipk + dMyc-RNAi #2*) (**h**) and *dMyc* knockdown disc (*dpp>GFP + dMyc-RNAi*) (**i**). GFP (green) marks the transgene-expressing cells. DAPI staining for DNA (blue) reveals tissue morphology. Scale bars, 50 μm. Exact *p* values are shown and calculated using unpaired two-tailed *t*-test.

*Figure 5 continued on next page*

*Figure 5 continued*

DOI: https://doi.org/10.7554/eLife.46315.013

The following source data is available for figure 5:

**Source data 1.** Analyses of glucose metabolism in *hipk*-expressing discs upon *dMyc* knockdown.

DOI: https://doi.org/10.7554/eLife.46315.014

(*Figure 5f*). Thus, while upregulation of dMyc is indispensable for Hipk-induced glycolysis and tumor growth, additional mechanisms are involved in the Hipk tumors to trigger the tumorigenic process.

## Aerobic glycolysis (the Warburg effect) is essential to tumor growth through sustaining ectopic dMyc expression

Earlier studies have shown that flies incompetent at glycolysis displayed reduced viability as deletion or ubiquitous knockdown of the glycolytic genes *pfk*, *ald*, *pgk*, *pglym78*, *eno* or *pyk* resulted in lethality (*Volkenhoff et al., 2015*; *Gerber et al., 2006*; *Miller et al., 2012b*; *Tennessen et al., 2011*). Likewise, we observed that flies ubiquitously depleted of *pgk* (*act5c>pgk-RNAi*) failed to reach the larval stage. Flies lacking *pfk* (*act5c>pfk-RNAi*) or *pyk* (*act5c>pyk-RNAi*) reached the larval stage but could not pupariate. Flies with *pfk2* knockdown (*act5c>pfk2-RNAi*) died at the early pupal stage but did not differentiate further to become late pupae. Most flies lacking *glut1* (*act5c>glut1-RNAi*) or *pgi* (*act5c>pgi-RNAi*) succeeded in emerging as adults. The timing of lethality following knockdown targeting glycolytic genes could presumably be influenced by maternal contribution, knockdown efficiency or the importance of the glycolytic step which the particular enzyme controls. To confirm the regulatory roles of Pfk and Pfk2 in glycolytic flux, we measured the relative pyruvate content in the knockdown larvae. Consistent with previous work (*Li et al., 2018*), knockdown of *pfk* significantly decreased pyruvate levels (*Figure 6—figure supplement 1*). Similarly, knockdown of *pfk2* induced an approximately 60% drop in the pyruvate content (*Figure 6—figure supplement 1*). The knockdown efficiencies of all RNAi constructs used in this study to silence glycolytic genes were evaluated by qRT-PCR analyses (*Figure 1—figure supplement 2*).

To examine the roles of glycolytic activation in Hipk tumor cells, we used RNAi to deplete key glycolytic genes. Knockdown of *hex-A* or *hex-C* had negligible effects on Hipk tumor growth; as did *hex-A* and *hex-C* double knockdown (*Figure 6—figure supplement 2*). Similarly, knockdown of *glut1* or *pgi* failed to block Hipk-induced tumorigenesis (*Figure 6—figure supplement 3a–b*). Knockdown of *pfk2*, on the contrary, markedly prevented tumor growth and restored tissue architecture in *hipk*-expressing discs (*Figure 6b*), suggesting that the committed step of glycolysis determines Hipk tumorigenesis. Indeed, the effect of *pfk* knockdown was comparable to that of *pfk2* knockdown (*Figure 6c*), even though we did not detect significant *pfk* upregulation (<1.5 fold) in *hipk*-expressing discs (*Figure 2a*). Knockdown of *pgk*, *pyk* or *Ldh* appeared to slightly restrict Hipk tumor growth but the tissue morphology remained distorted (*Figure 6d*, *Figure 6—figure supplement 3c–d*).

We also examined dMyc expression in Hipk tumor cells when glycolysis was inhibited. Intriguingly, dMyc accumulation was largely abolished in *hipk*-expressing discs when *pfk2* or *pfk* was knocked down (*Figure 6a–c*). In the absence of *hipk* overexpression, neither knockdown of *pfk* nor *pfk2* affected the normal dMyc expression pattern nor led to defects in tissue growth (*Figure 6—figure supplement 4*). Knockdown of *glut1*, *pgi*, *pgk*, *pyk* or *Ldh* in Hipk tumor cells did not rescue dMyc upregulation (*Figure 6d*, *Figure 6—figure supplement 3*). Together, our results suggest that dMyc accumulation and the associated Hipk tumor growth are most sensitive to the committed step of glycolysis governed by Pfk2-Pfk.

Using the *dMyc-lacZ* reporter, we found that *dMyc* transcriptional activation remained robust when *pfk2* or *pfk* was depleted in Hipk tumor cells even though the tumor growth was reduced and dMyc protein levels were reduced (*Figure 6e–g*). Thus, our data indicate that Pfk2 and Pfk are required to sustain ectopic dMyc expression specifically in Hipk tumor cells through post-transcriptional regulation.

The control of Hipk tumor growth by Pfk2 or Pfk seemed not to be a consequence of altered Hipk protein stability or functions because of the following reasons. First, we did not detect a noticeable change in Hipk protein levels in discs depleted of *pfk2* or *pfk* when compared with discs

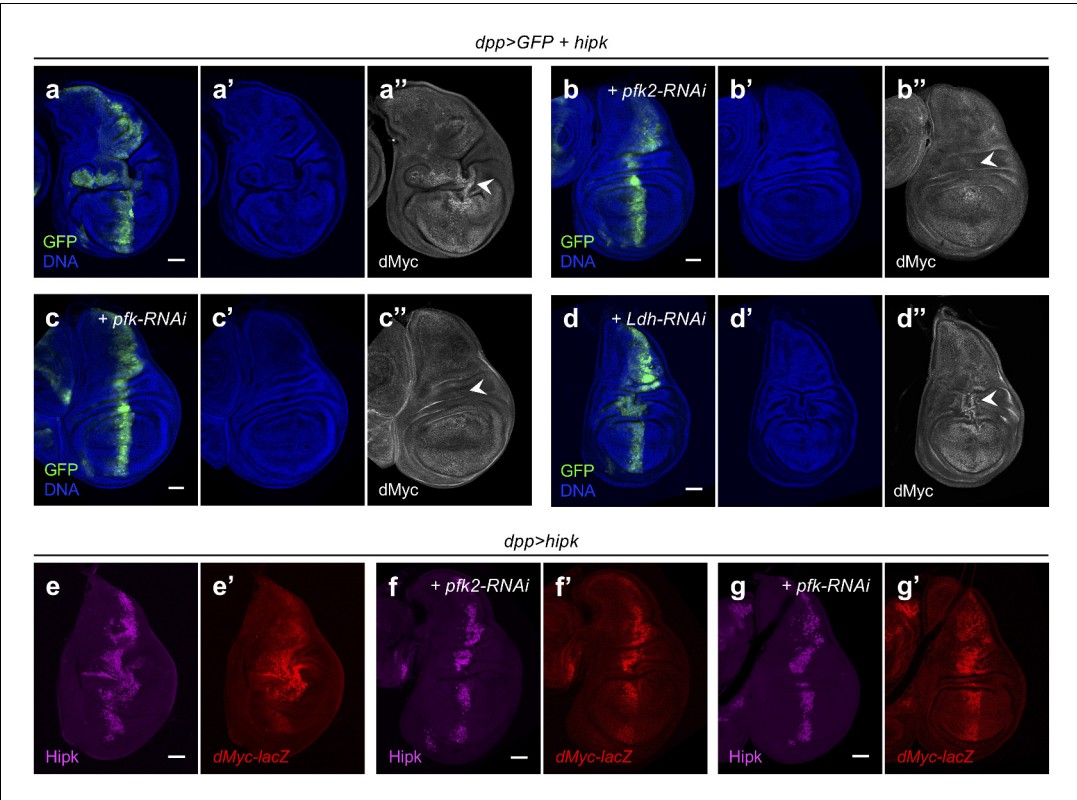

**Figure 6.** Pfk2 and Pfk sustain Hipk tumor growth through post-transcriptional regulation of dMyc. (**a–d**) dMyc staining (gray in **a''**, **b''**, **c''** and **d''**) in *hipk*-expressing wing disc (*dpp>GFP + hipk*) (**a**), *hipk* and *pfk2-RNAi* co-expressing disc (*dpp>GFP + hipk + pfk2-RNAi*) (**b**), *hipk* and *pfk-RNAi* co-expressing disc (*dpp>GFP + hipk + pfk-RNAi*) (**c**) and *hipk* and *Ldh-RNAi* co-expressing disc (*dpp>GFP + hipk + Ldh-RNAi*) (**d**). GFP (green) marks the transgene-expressing cells. DAPI staining for DNA (blue) reveals tissue morphology. (**e–g**) Hipk staining (magenta in **e**, **f** and **g**) and β-galactosidase staining (red in **e'**, **f'** and **g'**) in *hipk*-expressing disc (*dpp>hipk*) (**e**), *hipk* and *pfk2-RNAi* co-expressing disc (*dpp>hipk + pfk2-RNAi*) (**f**), *hipk* and *pfk-RNAi* co-expressing disc (*dpp>hipk + pfk-RNAi*) (**g**). All wing discs harbor a *lacZ* reporter to monitor the transcriptional induction of *dMyc* (*dMyc-lacZ*). Scale bars, 50 µm.

DOI: https://doi.org/10.7554/eLife.46315.015

The following source data and figure supplements are available for figure 6:

**Source data 1.** Data for pyruvate measurements.
DOI: https://doi.org/10.7554/eLife.46315.021
**Figure supplement 1.** Larvae depleted of *pfk2* or *pfk* display decreases in the pyruvate levels.
DOI: https://doi.org/10.7554/eLife.46315.016
**Figure supplement 2.** Hexokinases are not required for Hipk tumor growth.
DOI: https://doi.org/10.7554/eLife.46315.017
**Figure supplement 3.** Non-targeted inhibition of glycolysis is not sufficient to prevent Hipk tumor growth or ectopic dMyc accumulation.
DOI: https://doi.org/10.7554/eLife.46315.018
**Figure supplement 4.** Pfk2 and Pfk do not affect dMyc expression in the absence of *hipk* overexpression.
DOI: https://doi.org/10.7554/eLife.46315.019
**Figure supplement 5.** Knockdown of *pfk2* or *pfk* does not affect Hipk functions.
DOI: https://doi.org/10.7554/eLife.46315.020

expressing *hipk* alone (*Figure 6e–g*), indicating that blocking glycolysis does not trigger Hipk protein degradation. Second, the robust stimulation of the *dMyc-lacZ* reporter in *hipk*-expressing discs with *pfk2* or *pfk* knockdown suggests that Hipk activities remain intact. Indeed, when we examined other signaling targets downstream of Hipk as readouts for Hipk activity, we found that Hipk-mediated stabilization of Armadillo (Wnt/Wg signaling), and induction of Cyclin E or *expanded-lacZ*

(Hippo signaling) remained unaffected by knockdown of *pfk2* or *pfk* in a Hipk overexpression background, indicating Hipk is still functional (*Figure 6—figure supplement 5*).

Altogether, our data suggest that while dMyc upregulation promotes glycolytic activities, the upregulation of Pfk2, along with subsequent F2,6-BP biosynthesis and Pfk activation, is required to foster the accumulation of dMyc proteins. In other words, a positive feedback loop between dMyc and aerobic glycolysis is formed, coordinating growth signals with metabolic states to reinforce tumor progression.

## Discussion

### dMyc-driven aerobic glycolysis in a hipk tumor model

Multiple approaches and tools have been developed to measure glycolysis including measurement of the extracellular acidification rate (ECAR), fluorescent probes, biosensors, fluorescent/colorimetric assays and mass spectrometry (*Bittner et al., 2010*; *TeSlaa and Teitell, 2014*; *Tennessen et al., 2014*). In the Hipk tumor model, the tumor cells are surrounded by wild-type cells. Our study primarily used fluorescence imaging such that we could examine changes in metabolite levels, gene and protein expression between tumor cells and the adjacent wild-type cells. Our work delineates the causes and significance of metabolic changes in tumor cells.

*Drosophila* tumor models frequently acquire metabolic changes, especially the Warburg effect. For instance, epidermal growth factor receptor (EGFR)-driven tumors, tumors with activated platelet-derived growth factor (PDGF)/vascular endothelial growth factor (VEGF) receptor (Pvr) and tumors with polarity loss (such as *scrib* or *dlg* mutant tumors) feature robust upregulation of *Ldh* among other glycolytic genes (*Wang et al., 2016*; *Eichenlaub et al., 2018*; *Bunker et al., 2015*). Depletion of *Ldh* was shown to reduce growth of EGFR-driven tumors but not tumors with polarity loss (*Eichenlaub et al., 2018*; *Bunker et al., 2015*). In $Ras^{V12}scrib^{-/-}$ tumors, elevated glucose uptake was evident, but its significance was not evaluated (*Katheder et al., 2017*). Similar to the previously described models, Hipk tumor cells exhibited elevated glucose metabolism (*Figures 1–2*). Metabolic reprogramming was driven by dMyc upregulation (*Figures 3–5*). Although the transcript levels of another glycolytic inducer *sima* remained unchanged (*Figure 3a*), we could not eliminate the possibility that Sima protein levels are altered in the tumor cells. Thus, whether Sima is involved in the tumor growth warrants further studies. Genetical inhibition of Pfk or Pfk2 was sufficient to block Hipk-induced tumorigenesis whereas depletion of Pgk, Pyk or Ldh at most slightly reduced tumor growth (*Figure 6*; *Figure 6—figure supplement 3*). Therefore, our data suggest that targeted but not generic inhibition of glycolysis is required to abrogate tumor growth.

### A feedback loop between dMyc and aerobic glycolysis as a metabolic vulnerability in tumor cells

Although MYC functions in cancer metabolism are widely recognized, MYC is generally considered 'undruggable' largely due to its nuclear localization, lack of an enzymatic 'active site', and indispensable roles during normal development (*Soucek and Evan, 2010*). Hence, limited therapeutic strategies have been developed, and identification of the 'druggable' regulatory proteins of MYC becomes critical.

Our study reveals two modes of dysregulation of endogenous dMyc specifically during Hipk-induced tumorigenesis. The first mode is transcriptional stimulation of dMyc likely as a consequence of convergence of multiple signaling outputs (*Figures 3–4*). In our previous study, we found that genetically targeting individual signaling cascades failed to restrain Hipk tumor growth (*Blaquiere et al., 2018*). This is likely due to the redundancy of signaling cascades in inducing *dMyc* upregulation. It is also interesting to note that dMyc upregulation and the associated tumor growth are most striking in the wing hinge (*Figures 3–4*). The pouch region, however, seems more refractory to such alterations. We also observed the lowest intracellular glucose levels (*Figure 1g–h*) and robust *Ldh* upregulation (*Figure 2d*) in the hinge area. dMyc-dependent glucose uptake (*Figure 5b–c*), on the other hand, was enhanced in both hinge and pouch regions even though dMyc protein buildup is undetectable in the wing pouch, suggesting that glucose uptake is particularly sensitive to changes in dMyc levels. Our data therefore imply that the Warburg effect and the tumor growth phenotype may be linked to the dMyc levels being induced. The region-specific susceptibility to

tumorigenic stimuli has been previously described and the hinge region was coined a 'tumor hot-spot' because of its unique epithelial cell architectures and high endogenous JAK/STAT signaling (*Tamori et al., 2016*). It is possible that such features in the hinge region also contribute to the sensitivity of dMyc upregulation by elevated Hipk and other signaling pathways. Further studies are required to verify this proposition.

The second mode of dMyc regulation is the metabolic control of dMyc protein levels by aerobic glycolysis. Specifically, we found that the rate-limiting enzymes Pfk2 and Pfk are required to perpetuate dMyc buildup in tumor cells in a post-transcriptional manner (*Figure 6*). Similar to the effects on tumor growth, while *pfk2/pfk* knockdown prevented dMyc accumulation, knockdown of other glycolytic enzymes had little effect on sustaining dMyc accumulation. Such a disparity could possibly be explained by the potency of the glycolytic enzymes in controlling glycolytic flux. While Pfk2 and Pfk govern the rate-limiting, committed step of glycolysis, RNAi targeting other glycolytic genes may fail to render the corresponding steps rate-limiting, especially in tumors with elevated aerobic glycolysis. In other words, knockdown of other glycolytic enzymes may not reach the threshold that would restrict glycolytic flux as potently as *pfk2/pfk* knockdown. Given that Pfk2 is mis-expressed in tumor cells but not in normal cells (*Figure 2a*) and that dMyc accumulation is most sensitive to the Pfk2-Pfk mediated committed step of glycolysis specifically in tumor cells (*Figure 6*), targeting Pfk2 may be a favorable, selective metabolic strategy in the treatment of cancers, especially those displaying ectopic MYC expression.

### Potential pro-tumorigenic roles of the pfk and Pfk2 in addition to their core biochemical role in glycolysis

The human homologs of fly Pfk and Pfk2 are PFK and PFK-2/FBPase (PFKFB), respectively. The functions of mammalian PFK and PFKFB in controlling glycolytic flux are well-defined. The allosteric regulation of PFK by F2,6-BP seems to be conserved among mammals and insects as F2,6-BP is able to activate insect Pfk, likely due to the conserved amino acid residues for F2,6-BP binding (*Nunes et al., 2016*). This suggests that Pfk2 can induce Pfk activation and hence glycolytic flux through biosynthesizing F2,6-BP. A previous report shows that flies lacking *pfk2* exhibited levels of circulating glucose and reduced sugar tolerance, providing a functional link between Pfk2 and glucose metabolism (*Havula et al., 2013*). We provide evidence that both *pfk2* and *pfk* knockdown larvae exhibited significant decreases in pyruvate contents, confirming a conserved role of Pfk2 in regulating glycolytic flux.

Recently, the roles of PFK and PFKFB in the regulation of transcription factors or cofactors have drawn considerable attention. For example, vertebrate PFK was shown to physically interact with TEAD factors to stimulate YAP/TAZ activity, thus promoting proliferation and malignancy in cancer cells (*Enzo et al., 2015*). PFKFB4 (an isoform of PFK2) phosphorylates and activates oncogenic SRC-3 to promote breast cancers through stimulating purine synthesis (*Dasgupta et al., 2018*). These studies point to the fact that PFK and PFKFB functions are not restricted to metabolic regulation. Thus, it is tempting to speculate that Pfk2/Pfk may sustain ectopic dMyc through a direct interaction with or even phosphorylation of dMyc. That being said, as knockdown of Pfk2 or Pfk had no effects on dMyc protein levels under normal conditions (*Figure 6—figure supplement 4*), we tend to believe that a tumorigenic environment with active aerobic glycolysis is necessary for the metabolic control of dMyc in particular through the committed step governed by Pfk2/Pfk. In summary, we and others show that glycolytic enzymes Pfk and Pfk2, likely through their core biochemical role in stimulating glycolytic flux, glycolysis-independent actions or both, act as key players in coupling metabolic demands with growth signals to achieve cancer progression.

## Materials and methods

### *Drosophila* genetics

*Drosophila melanogaster* flies were raised on standard cornmeal-molasses food. Stocks were kept at 25°C and crosses at 29°C unless otherwise indicated. Three Gal4 fly lines, (1) *dpp-Gal4/TM6B*, (2) *hh-Gal4/TM6B* and (3) *act5c-Gal4/TM6B* (BL #3954) were used to induce transgene expression. To generate heat-shock inducible actin flip-out clones, (4) *yw,hsFlp[122];;act>CD2>Gal4, UAS-RFP/TM6B* was used. To mark the transgene-expressing cells, (5) *UAS-GFP* (BL #5431) and (6) *UAS-RFP* (BL

#7118) were used. To generate the Hipk tumor model, (7) *UAS-HA-hipk-3M* (abbreviated as *UAS-hipk*) (*Swarup and Verheyen, 2011*) and (8) *dpp-Gal4, UAS-hipk/TM6B* were used. RNAi fly strains used include (9) *UAS-dMyc-RNAi* (BL #25783), (10) *UAS-dMyc-RNAi #2* (BL #25784), (11) *UAS-glut1-RNAi* (BL #40904), (12) *UAS-hex-A-RNAi* (BL #35155), (13) *UAS-hex-C-RNAi* (BL #57404), (14) *UAS-pfk2-RNAi* (BL #35380), (15) *UAS-pfk-RNAi* (BL #34336), (16) *UAS-Ldh-RNAi* (BL #33640), (17) *UAS-pgi-RNAi* (BL #51804), (18) *UAS-pgk-RNAi* (BL #33632) and (19) *UAS-pyk-RNAi* (BL #35218). To perturb signaling pathway activities, the following strains were used: (20) *UAS-yki-S168A-GFP* (BL #28836), (21) *UAS-Arm-S10* (*Mirkovic et al., 2011*), (22) *UAS-Ci-5M*, (23) *UAS-N-act*. Other strains used include (24) *myc-lacZ/FM7c* (BL #12247) and (25) *UAS-dMyc* (BL #9674). (26) *UAS-FLII12Pglu-700μδ6* (Glucose sensor) is a generous gift from Dr. Stefanie Schirmeier (*Volkenhoff et al., 2018*). (27) *Ldh-GFP* is a generous gift from Dr. Ingrid Lohmann. Strains with BL stock number were obtained from Bloomington *Drosophila* Stock Center (Bloomington, IN, USA).

## Flip-out clones
A 10–15 min heat shock (37°C) was applied to larvae grown for two to three days after egg laying (25°C). After heat shock, the larvae were grown at 29°C and dissected when they reached third instar larval stage.

## Immunofluorescence microscopy
Larval imaginal discs were dissected in PBS and fixed in 4% paraformaldehyde (PFA) for 15 min at room temperature. Samples were washed with PBS with 0.1% Triton X-100 (PBST). After blocking with 5% BSA in PBST for 1 hr at room temperature, samples were incubated with primary antibodies overnight at 4°C. The following primary antibodies were used: (1) mouse anti-β-Galactosidase (1:50, DSHB 40-1a), (2) rabbit anti-dMyc (d1-717) (1:500, Santa Cruz Biotechnology, Inc sc-28207), (3) rabbit anti-Hipk (1:200; *Blaquiere et al., 2018*), (4) mouse anti-Armadillo (1:50, diluted form, DSHB N2 7A1), (5) rabbit anti-Cyclin E (d-300) (1:100, Santa Cruz Biotechnology, Inc sc-33748). After washing with PBST, samples were incubated with Cy3- and/or Alexa Fluor 647-conjugated secondary antibodies (1:500, Jackson ImmunoResearch Laboratories, Inc), DAPI (4',6-Diamidino-2-Phenylindole, Dihydrochloride) (final concentration: 0.2 μg per mL, Invitrogen D1306) and/or Rhodamine phalloidin (1:500, Molecular Probes R415) for 2 hr at room temperature. Samples were mounted in 70% Glycerol/PBS after wash. Images were taken on a Nikon Air laser-scanning confocal microscope and processed by Image J.

## Quantitative RT-PCR (qRT-PCR)
Imaginal discs from third instar larva were dissected in PBS and temporarily stored in RNAlater Stabilization solution (Invitrogen AM7020). Total RNA was extracted using RNeasy Mini Kits (Qiagen 74101). First strand cDNA was synthesized using OneScript Plus cDNA Synthesis Kit (Abm G236). qRT-PCR were performed using SensiFast SYBR Hi-ROS Kit (Bioline 92005) on StepOne Real-time PCR System (Applied Biosystems).

## 2-NBDG uptake
Imaginal discs from third instar larva were dissected in PBS, followed by incubation with 2-NBDG (Invitrogen N13195) at 0.3 mM for 15 min in the dark at room temperature. After rinsing with PBS, samples were fixed and processed according to standard immunofluorescence staining protocol.

## Intracellular glucose FRET analysis
The glucose sensor *UAS-FLII*[12]*Pglu-700μδ6* was generated by Dr. Schirmeier's group (*Volkenhoff et al., 2018*). Imaginal discs expressing the sensor with or without overexpression of *hipk* were mounted and imaged on the same day. The same confocal settings as previously described (*Volkenhoff et al., 2018*) were used to record the CFP and FRET images. FRET efficiency (FRET/CFP) was calculated as quotient of FRET and CFP using Image J. Since we were not determining the actual values of intracellular glucose concentrations, we did not take into account a correction for spectral bleed through and thus uncorrected FRET/CFP ratios were used to compare glucose levels in discs of different genotypes. The values of FRET efficiency were displayed using the Fire Lookup Table (LUT) in figures and shown in boxplots.

## Pyruvate measurement

Whole-body larvae were homogenized in PBS, followed by centrifugation at 14,000 g for 10 min. The supernatants were used for the pyruvate assay. The pyruvate assay was performed using Enzy-Chrom Pyruvate Assay Kit (BioAssay Systems EPYR-100) according to the manufacturer's instructions. Absorbance at 570 nm was measured in a SpectraMax M5 fluorescent microplate reader (Molecular Devices). Relative pyruvate levels were determined after normalizing with the protein levels. Protein quantification was performed using the Bio-Rad protein assay (Bio-Rad 500–0006) and absorbance at 595 nm was measured. Experiments were conducted with two biological replicates. Each assay was performed with two technical replicates.

## Statistics

For data analyses, unpaired two-tailed Student t-tests were used to determine $p$-values using Microsoft Excel. Bar graphs were generated by Microsoft Excel. Boxplots were generated using BoxPlotR with data points included (*Spitzer et al., 2014*). We used the Spear definition to show the maximum and minimum values by the upper and lower whiskers, respectively. The third quartile, median and the first quartile are shown in the box accordingly to a standard boxplot.

# Acknowledgements

We are grateful to the Developmental Studies Hybridoma Bank, the Bloomington *Drosophila* Stock Center and Jackson ImmunoResearch Laboratories for providing fly strains and antibodies. We thank Dr. S Schirmeier for providing the glucose sensor *UAS-FLII$^{12}$Pglu-700μδ6* flies. We thank Dr. I Lohmann for providing the *Ldh-GFP* flies. We appreciate all the discussion and inputs from our laboratory members including Dr. D Sinclair, Dr. G Tettweiler, S Kinsey, L Madonsela, R Tse and T Wiens. This work was funded by an operating grant from the Canadian Institutes of Health Research, PJT-156204.

# Additional information

### Funding

| Funder | Grant reference number | Author |
| --- | --- | --- |
| Canadian Institutes of Health Research | PJT-156204 | Esther M Verheyen |

The funders had no role in study design, data collection and interpretation, or the decision to submit the work for publication.

### Author contributions

Kenneth Kin Lam Wong, Conceptualization, Data curation, Formal analysis, Investigation, Visualization, Writing—original draft, Writing—review and editing; Jenny Zhe Liao, Formal analysis, Investigation, Visualization, Writing—review and editing; Esther M Verheyen, Conceptualization, Formal analysis, Supervision, Funding acquisition, Project administration, Writing—review and editing

### Author ORCIDs

Esther M Verheyen https://orcid.org/0000-0002-9795-5094

### Decision letter and Author response

Decision letter https://doi.org/10.7554/eLife.46315.025
Author response https://doi.org/10.7554/eLife.46315.026

## Additional files

### Supplementary files

• Supplementary file 1. A list of *Drosophila* genes studied in this work with their human homologs.
DOI: https://doi.org/10.7554/eLife.46315.022

• Transparent reporting form
DOI: https://doi.org/10.7554/eLife.46315.023

### Data availability

Data generated or analyzed during this study are available in the manuscript and supporting files.

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
