## [Decision Letter]

Thank you for submitting your article "Metabolic alterations involving a dMyc-Pfk2-Pfk regulatory loop and active mitochondrial metabolism drive tumor growth" for consideration by *eLife*. Your article has been reviewed by three peer reviewers, including Jason Tennessen as the guest Reviewing Editor and Reviewer #1, and the evaluation has been overseen by Utpal Banerjee as the Senior Editor.

The reviewers have discussed the reviews with one another and the Reviewing Editor has drafted this decision to help you prepare a revised submission.

Summary:

The consensus view is that the manuscript provides an interesting description of a mechanism linking Hipk with aerobic glycolysis and mitochondrial metabolism within *Drosophila* larval cells. The manuscript provides three important observations:

1) Hipk overexpression within the wing imaginal disc induces elevated glycolytic metabolism as determined by 2-NBDG uptake, a glucose FRET-sensor, and changes in gene expression related to glycolysis. Overall, these findings suggest that Hipk overexpression induces the Warburg effect in the wing disc.

2) Hipk overexpression within the larval salivary glands induces significant changes in mitochondrial morphology and membrane potential.

3) Loss of either Pfk or Pfk2 expression results in a decreased dMyc levels, hinting at a mechanism in which Hipk induced expression of Pfk and Pfk2 promotes dMyc accumulation.

Individually, each of these points is interesting, however, the studies of mitochondrial metabolism are largely conducted in the salivary gland while the glycolytic studies primarily take place within the wing imaginal disc. Combining these two tissue-specific studies into a single manuscript raises the question as to whether Hipk activates different metabolic programs within each tissue. This dilemma is further emphasized by the last paragraph of the Results section, which revealed that PFK2 but not PFK is required for mitochondrial elongation and branching phenotypes observed in the salivary glands. Since this result is in contrast to the Hipk-induced imaginal disc tumors, which are dependent on both PFK and PFK2, substantial effort would be required to understand how Hipk differentially regulates metabolism in these two tissues.

After several discussions, the reviewers agree that the strength of this study lies in the relationship between Hipk, dMyc, and Pfk/Pfk2 within the wing imaginal disc. The mechanism proposed by the wing disc studies are better defined than the salivary gland work and they reflect the heart of the manuscript. Based on this conclusion, the manuscript should be split in two, with the wing disc studies serving as the foundation of a major revision and the mitochondrial studies acting as seed for a future manuscript.

Essential revisions:

The reviewers agree that the following major revisions are required before the paper can be accepted at *eLife*:

1) Reorganize the manuscript to focus on the link between Hipk, Myc, and glycolysis in the imaginal disc while removing the mitochondrial studies in the salivary gland.

2) The reviewers request three additional sets of experiments to demonstrate that glycolysis is both elevated and necessary in HIPK-induced tumors.

i) The reviewers would like additional experiments demonstrating that 2-NBDG serves as an accurate readout of cellular glucose import:

How efficient is the UAS-Glut-RNAi used?

Is Glut1 over-expression sufficient to cause an accumulation in 2-NBDG?

Does Glut1 depletion rescue tumor formation?

ii) Please provide some evidence that Pfk2 activity is linked with increased glycolytic flux. Insect PFK is regulated differently than mammalian PFK, as the insect enzymes are insensitive to both citrate and AMP levels. This evidence could be provided by an appropriate set of references from the insect metabolism literature, a biochemical study using S2 cells, or measurements of lactate and pyruvate in larvae following whole body Pfk-2-RNAi.

iii) The reviewers are concerned that the link between Pfk/Pfk2 and dMyc might not be the result of a direct interaction, but instead reflects a feedback from glycolysis onto dMyc stability. They request that the authors use RNAi to determine if depletion of other glycolytic enzymes affects Myc levels. Specifically, the authors should determine if dMyc levels are sensitive to the loss of a glycolytic enzyme that is encoded by a single gene (unlike the hexokinases) and directly catalyzes a glycolytic reaction (unlike Ldh). Possible suggestions would be Pgi, Pyk, or Pgk.

3) The authors must strengthen the connection between HIPK up-regulation and dMyc expression.

i) The authors claim in Figure 4 that HIPK induces elevated dMyc levels. While the increase in dMyc-lacZ resembles the dpp-expression pattern, dMyc protein levels exhibit a different pattern. The authors should clarify this discrepancy. Moreover, dMyc protein levels appear to be increased in some regions of the disc but not in all the dpp-expressing cells (e.g., the wing pouch in Figure 4E, or Figure 6A). The use of clones over-expressing Hipk could help the authors to assess this potential region specificity in dMyc protein upregulation.

ii) There are also concerns regarding the section that examines the role of different signaling pathways on dMyc protein levels. The authors examine dMyc protein when different pathways are deregulated in the disc but do not evaluate the effect of dMyc expression. The use of clones might give better resolution and will allow showing these results in a clearer manner.

[Editors' note: further revisions were requested prior to acceptance, as described below.]

Thank you for resubmitting your work entitled "A dMyc-Pfk2-Pfk regulatory loop drives the Warburg effect and tumor growth" for further consideration at *eLife*. Your revised article has been favorably evaluated by Utpal Banerjee (Senior Editor), a Reviewing Editor, and three reviewers.

The manuscript has been substantially improved but there are a few remaining issues that need to be addressed before acceptance. As outlined by reviewer 3 and mentioned in my comments (reviewer 1), there are still concerns regarding the conclusion that PfK and Pfk2 promote dMyc stability independent of glycolytic flux. I would ask that you address these concerns within the text. In particular, please discuss alternative models that could explain your observations and justify why you favor a glycolysis-independent role for Pfk and Pfk2.

*Reviewer #1:*

The revised manuscript Wong, Lio, and Verheyen is significantly improved and all of my major concerns have been addressed. I have two minor revisions that the authors should address prior to publication.

1) In subsection “Aerobic glycolysis (the Warburg effect) is essential to tumor growth through sustaining ectopic dMyc expression”, the statement that glycolysis is indispensable for *Drosophila* growth and development needs some citations. Also, the authors should revise the sentence to state that glycolysis is indispensable for larval development….Pfk mutants can survive until L2 and therefore survive embryogenesis. While this is likely due to maternal loading of the enzyme, the authors should use a more nuanced statement to start this section.

2) The glycolytic studies are significantly improved. But the authors can't rule out the possibility that RNAi targeting the other genes, such as Pgk and Pgi, fails to lower enzyme activity below a level that would restrict glycolytic flux. A brief qualifying statement about this in the discussion would alleviate my concerns.

*Reviewer #2:*

In the revised version of their manuscript, the authors have addressed all my concerns and, in my view, it is ready for publication.

*Reviewer #3:*

The manuscript is improved following the omission of the salivary gland data. The focus on the wing disc eliminates the confusion caused by studying the divergent phenotypes in the different tissues.

Much of the manuscript now discusses the role of glycolysis in tumor growth. The authors were asked to provide additional data to bolster the assertion that Hipk over-expression causes glycolysis to be increased and that the increase in glycolysis is necessary. Particularly, the authors were also asked to demonstrate that the stability of dMyc was due to a direct interaction between Pfk/Pfk2 and dMyc.

While the authors performed many of the experiments requested, it is still not clear if the changes in tumor growth and dMyc stability are due to a direct interaction between Pfk or Pfk2 with dMyc (i.e. a mechanism that does not involve glycolysis) or whether the glycolytic role of Pfk is critical, but glycolysis is especially sensitive to inhibition at the Pfk step. In the rebuttal, the authors propose that the direct interaction model is supported by negative data describing the relative unimportance of the other glycolytic steps. The authors performed RNAi knock-downs of the other glycolytic enzymes (as requested during the initial review). These experiments show that knock-down of the other glycolytic enzymes do not reduce dMyc stability, but have also only, surprisingly, a very minor impact on tumor growth (Figure 6—figure supplement 1, Figure 6—figure supplement 2, Figure 6—figure supplement 3, Figure 6—figure supplement 4 and Figure 6—figure supplement 5). Is it really true, then, that elevated glucose import or glycolysis is not necessary for tumor growth, or could it be that the knock-downs of the other glycolytic enzymes do not reduce glycolytic flux to the same extent as the Pfk/Pfk2 knock-down? This is a very difficult question to resolve without quantitative flux data, though accepting the conclusion that elevated glycolysis is not necessary for tumor growth would be contrary to much previous work. The other solution to distinguishing between the two models would be to provide a clear mechanistic model that describes how Pfk/Pfk2 promotes dMyc stability independent of the role of Pfk in glycolysis – the authors provide no new data to describe how such a mechanism might operate.

In the absence of new data, I don't think a glycolysis-independent role of Pfk is supported – right now it seems very difficult to separate the two phenomena. Any model that is proposed should take this uncertainty more fully into account. On the other hand, it seems very reasonable to say the targeted inhibition of glycolysis is necessary to reduce tumor-growth and that the Pfk/Pfk2 step is the most sensitive step in this system (e.g. subsection “dMyc-driven metabolic rewiring in Hipk tumor model”).

---

## [Author Response]

Essential revisions:The reviewers agree that the following major revisions are required before the paper can be accepted at eLife:1) Reorganize the manuscript to focus on the link between Hipk, Myc, and glycolysis in the imaginal disc while removing the mitochondrial studies in the salivary gland.

Our manuscript now focuses on the link the between Hipk tumor growth, dMyc upregulation and the Warburg effect in the wing imaginal discs.

2) The reviewers request three additional sets of experiments to demonstrate that glycolysis is both elevated and necessary in HIPK-induced tumors.i) The reviewers would like additional experiments demonstrating that 2-NBDG serves as an accurate readout of cellular glucose import:How efficient is the UAS-Glut-RNAi used?

The knockdown efficiency of the *UAS-glut1-RNAi* was evaluated by qRT-PCR. We found that expression of *glut1-RNAi* led to a ~70% decrease in *glut1* mRNA expression. Other RNAi lines used in this study to silence glycolytic genes were also tested. These data are shown in Figure 1 —figure supplement 2.

Is Glut1 over-expression sufficient to cause an accumulation in 2-NBDG?

It has been shown that expression of *glut1* is sufficient to increase cellular uptake of 2-NBDG (Niccoli et al.,2016), demonstrating that 2-NBDG is a valid tool to monitor glucose uptake in *Drosophila*. Additional examples of using 2-NBDG in *Drosophila* are provided in the manuscript. This information is included in the second paragraph of the Results section.

Does Glut1 depletion rescue tumor formation?

We found that depletion of *glut1* failed to rescue Hipk tumor growth (Figure 6—figure supplement 3) despite our results showing that knockdown of *glut1* moderately reduces glucose transport in the tumor cells. We also checked the tumor growth when we knocked down those glycolytic enzymes without genetic redundancy. Depletion of Pgk, Pyk or Ldh at most slightly reduced tumor growth, whereas Pfk or Pfk2 knockdown blocked tumor growth. This implies that whereas aerobic glycolysis might partly fuel overgrowth, targeted but not generic inhibition of glycolysis is required to abrogate tumor growth. See more in the section below – 2 iii.

ii) Please provide some evidence that Pfk2 activity is linked with increased glycolytic flux. Insect PFK is regulated differently than mammalian PFK, as the insect enzymes are insensitive to both citrate and AMP levels. This evidence could be provided by an appropriate set of references from the insect metabolism literature, a biochemical study using S2 cells, or measurements of lactate and pyruvate in larvae following whole body Pfk-2-RNAi.

Nunes et al.,(2016) reported that unlike mammalian PFK, PFK from insects (*A. aegypti*) is insensitive to citrate or AMP likely due to the non-conserved amino acid sequences for citrate and AMP binding sites. Interestingly, in the study, the authors found that F2,6-BP is able to activate insect PFK, suggesting that the regulation of PFK by F2,6-BP is conserved among mammals and insects. Indeed, the amino acid residues involved in F2,6-BP binding in PFK are conserved among mammals and insects including *Drosophila*.

The roles of Pfk2 in *Drosophila* are less studied. We only found one study showing that flies depleted of Pfk2 exhibit elevated levels of circulating glucose and reduced sugar tolerance (Havula et al., 2013). In our study, we found that knockdown of *pfk2* mimics the effects of knockdown of other key glycolytic genes, causing lethality in flies. Also, *pfk2* knockdown phenocopies the effect of *pfk* knockdown, leading to decreases in pyruvate levels in flies (Figure 6—figure supplement 1).

All these pieces of evidence illustrate the importance of fly Pfk2 in controlling glycolytic flux.

iii) The reviewers are concerned that the link between Pfk/Pfk2 and dMyc might not be the result of a direct interaction, but instead reflects a feedback from glycolysis onto dMyc stability. They request that the authors use RNAi to determine if depletion of other glycolytic enzymes affects Myc levels. Specifically, the authors should determine if dMyc levels are sensitive to the loss of a glycolytic enzyme that is encoded by a single gene (unlike the hexokinases) and directly catalyzes a glycolytic reaction (unlike Ldh). Possible suggestions would be Pgi, Pyk, or Pgk.

To address whether elevated glycolytic activities contribute to the maintenance of dMyc expression, we used RNAi to knockdown *glut1, pgi, pyk* and *pgk*. Unlike *pfk2/pfk* knockdown, we found that none of the knockdowns suppressed dMyc upregulation (Figure 6—figure supplement 3), suggesting that loss of dMyc in *pfk2/pfk* knockdown is not due to attenuation of glycolytic flux.

3) The authors must strengthen the connection between HIPK up-regulation and dMyc expression.i) The authors claim in Figure 4 that HIPK induces elevated dMyc levels. While the increase in dMyc-lacZ resembles the dpp-expression pattern, dMyc protein levels exhibit a different pattern. The authors should clarify this discrepancy. Moreover, dMyc protein levels appear to be increased in some regions of the disc but not in all the dpp-expressing cells (e.g., the wing pouch in Figure 4E, or Figure 6A). The use of clones over-expressing Hipk could help the authors to assess this potential region specificity in dMyc protein upregulation.

We examined dMyc upregulation in *dpp>hipk* discs where *hipk* was overexpressed along the anterior/posterior boundary. We observed that the increase in the β-gal signal appeared to be the largest in the presumptive wing hinge where the tumor cells proliferated most rapidly, while the intensity decreased when moving away from the hinge region. We detected accumulation of dMyc proteins in the wing hinge, which positively correlates with the marked increase in dMyc mRNA levels in the same area. On the other hand, dMyc protein levels remained unchanged in the wing pouch, possibly due to weak induction of dMyc transcription

ii) There are also concerns regarding the section that examines the role of different signaling pathways on dMyc protein levels. The authors examine dMyc protein when different pathways are deregulated in the disc but do not evaluate the effect of dMyc expression. The use of clones might give better resolution and will allow showing these results in a clearer manner.

Using the strategy mentioned above, we used clones to examine the roles of different signaling pathways on dMyc expression. Similar to elevated Hipk, we observed that perturbations in the tested signaling cascades cause ectopic dMyc expression specifically in the wing hinge but not in the pouch region. Data are shown in Figure 4 and Figure 4—figure supplement 1. We discussed such dichotomy in the Discussion section.

[Editors' note: further revisions were requested prior to acceptance, as described below.]

The manuscript has been substantially improved but there are a few remaining issues that need to be addressed before acceptance. As outlined by reviewer 3 and mentioned in my comments (reviewer 1), there are still concerns regarding the conclusion that PfK and Pfk2 promote dMyc stability independent of glycolytic flux. I would ask that you address these concerns within the text. In particular, please discuss alternative models that could explain your observations and justify why you favor a glycolysis-independent role for Pfk and Pfk2.

We thank the editors and reviewers for their thoughtful and constructive feedback to our revised manuscript. We believe that in this edited version, we have addressed the concerns and discussed broader interpretations, and we hope the reviewers agree.

Given that inhibition of Pfk2 and Pfk potently rescues dMyc accumulation and the associated tumor growth in our Hipk tumor model, we previously focused on the potential links between the Pfk2-Pfk axis and dMyc, which could be therapeutically valuable in cancer treatment. We agree with reviewers that the suppression effect could be explained by two alternative models, a direct interaction between dMyc and Pfk2/Pfk and a general attenuation of glycolytic flux. In light of this, we changed the focus from the dMyc-Pfk2-Pfk interactions to emphasizing the feedback loop between dMyc and aerobic glycolysis, and correspondingly have changed the title to “A positive feedback loop between Myc and aerobic glycolysis sustains tumor growth in a *Drosophila* tumor model”. Furthermore, we highlight that inhibition of the Pfk2/Pfk-governed committed step of glycolysis may serve as an effective approach to abrogate tumor growth.

In the last paragraph of the Discussion section, we provide some references showing that in human cancer cells, PFK and PFKFB (Pfk2 in flies) can physically interact or directly phosphorylate oncogenic proteins like TEAD factors (Enzo et al.,2015) or SRC-3 (Dasgupta et al.,2018). Thus, it is tempting to speculate that Pfk2/Pfk may sustain ectopic dMyc through a direct interaction with or even phosphorylation of dMyc (subsection “Potential pro-tumorigenic roles of the Pfk and Pfk2 in addition to their core biochemical role in glycolysis”). However, we currently do not have appropriate reagents like fly strains or antibodies to investigate the potential physical interactions, and those studies are beyond the scope of the current study.

In summary, we and others show that glycolytic enzymes Pfk and Pfk2, likely through their core biochemical role in stimulating glycolytic flux, glycolysis-independent actions or both, act as key players in coupling metabolic demands with growth signals to achieve cancer progression. (subsection “Potential pro-tumorigenic roles of the Pfk and Pfk2 in addition to their core biochemical role in glycolysis”)

Reviewer #1:The revised manuscript Wong, Lio, and Verheyen is significantly improved and all of my major concerns have been addressed. I have two minor revisions that the authors should address prior to publication.

*1) In subsection “Aerobic glycolysis (the Warburg effect) is essential to tumor growth through sustaining ectopic dMyc expression”, the statement that glycolysis is indispensable for Drosophila growth and development needs some citations. Also, the authors should revise the sentence to state that glycolysis is indispensable for larval development….Pfk mutants can survive until L2 and therefore survive embryogenesis. While this is likely due to maternal loading of the enzyme, the authors should use a more nuanced statement to start this section.*

We have replaced the statement by ‘Earlier studies have shown that flies incompetent at glycolysis displayed reduced viability’. We also refer to previous studies that showed that deletion or ubiquitous knockdown of glycolysis genes resulted in lethality in files, which are consistent with our knockdown analyses.

We also mention that the lethal phenotype could presumably be determined by maternal contribution, knockdown efficiency, or the importance of the glycolytic step which the particular enzyme controls.

2) The glycolytic studies are significantly improved. But the authors can't rule out the possibility that RNAi targeting the other genes, such as Pgk and Pgi, fails to lower enzyme activity below a level that would restrict glycolytic flux. A brief qualifying statement about this in the discussion would alleviate my concerns.

We now provide the required statements when we discuss the different effects of Pfk2/Pfk knockdown or knockdown of other glycolytic genes on tumor growth and dMyc accumulation.

Reviewer #3:

[…] In the absence of new data, I don't think a glycolysis-independent role of Pfk is supported – right now it seems very difficult to separate the two phenomena. Any model that is proposed should take this uncertainty more fully into account. On the other hand, it seems very reasonable to say the targeted inhibition of glycolysis is necessary to reduce tumor-growth and that the Pfk/Pfk2 step is the most sensitive step in this system (e.g. subsection “dMyc-driven metabolic rewiring in Hipk tumor model”).

Some of these concerns have been addressed in our response to reviewer 1.

In addition, we are inclined to believe that aerobic glycolysis is crucial for tumor growth despite a potential direct interaction between dMyc and Pfk2/Pfk. We found that knockdown of Pfk2 or Pfk had no effects on dMyc protein levels under normal conditions (Figure 6—figure supplement 4). This suggests that a tumorigenic environment with active aerobic glycolysis is essential for the metabolic control of dMyc in particular through the committed step governed by Pfk2/Pfk. (subsection “Potential pro-tumorigenic roles of the Pfk and Pfk2 in addition to their core biochemical role in glycolysis).